# IMPLICIT BIAS OF LINEAR EQUIVARIANT NETWORKS

## ABSTRACT

Group equivariant convolutional neural networks (G-CNNs) are generalizations of convolutional neural networks (CNNs) which excel in a wide range of scientific and technical applications by explicitly encoding group symmetries, such as rotations and permutations, in their architectures. Although the success of G-CNNs is driven by the *explicit* symmetry bias of their convolutional architecture, a recent line of work has proposed that the *implicit* bias of training algorithms on a particular parameterization (or architecture) is key to understanding generalization for overparameterized neural nets. In this context, we show that $L$-layer full-width linear G-CNNs trained via gradient descent in a binary classification task converge to solutions with low-rank Fourier matrix coefficients, regularized by the $2/L$-Schatten matrix norm. Our work strictly generalizes previous analysis on the implicit bias of linear CNNs to linear G-CNNs over all finite groups, including the challenging setting of non-commutative symmetry groups (such as permutations), as well as band-limited G-CNNs over infinite groups. We validate our theorems via experiments on a variety of groups and empirically explore more realistic nonlinear networks, which locally capture similar regularization patterns. Finally, we provide intuitive interpretations of our Fourier space implicit regularization results in real space via uncertainty principles.

## 1 INTRODUCTION

Modern deep learning algorithms typically have many more parameters than data points, and their ability to achieve good generalization in this overparameterized setting is largely unexplained by current theory. Classic generalization bounds, which bound generalization errors when models are not overly "complex," are vacuous for neural networks that can perfectly fit random training labels (Zhang et al., 2017). More recent work analyzes the complexity of deep learning algorithms by instead characterizing the properties of the functions they output. Notably, prior work has shown that training via gradient descent implicitly regularizes towards certain hypothesis classes with low complexity, which may generalize better as a result. For example, in underdetermined least squares regression, gradient descent converges to the $\ell_2$-norm minimizer, while a pointwise-square reparametrization converges to the $\ell_1$-norm minimizer (Gunasekar et al., 2018a). Such phenomena are consistent with certain linear neural networks, *e.g.,* Gunasekar et al. (2018b) showed that learned linear fully-connected and convolutional networks implicitly regularize the $\ell_2$ norm and a depth-dependent norm in the Fourier space respectively.

From a more applied perspective, a large body of work imposes structured inductive biases on deep learning algorithms to exploit symmetry patterns (Kondor, 2007; Reisert, 2008; Cohen & Welling, 2016b). One prominent method parameterizes models over functions that are *equivariant* with respect to a symmetry group (*i.e.,* outputs transform predictably in response to input transformation). In fact, Kondor & Trivedi (2018) showed that any group equivariant network can be expressed as a series of group convolutional layers interwoven with pointwise nonlinearities, demonstrating that group convolutional neural networks (G-CNNs) are the most general family of equivariant networks.

**Our Contributions:** The explicit inductive bias of G-CNNs is the main reason for their usage. Yet, to the best of our knowledge, the implicit bias imposed by equivariant architectures has not been explored. Here, we greatly generalize the results of Yun et al. (2020) and Gunasekar et al. (2018b) to linear G-CNNs whose hidden layers perform group equivariant transformations. We show, surprisingly, that $L$-layer G-CNNs are implicitly regularized by the $2/L$-Schatten norm, which is the $2/L$ norm of a matrix's singular values, over the irreducible representations in the Fourier basis. As

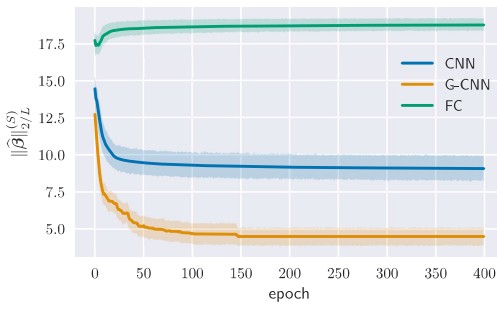

(a) Fourier space norm of network linearization $\boldsymbol{\beta}$

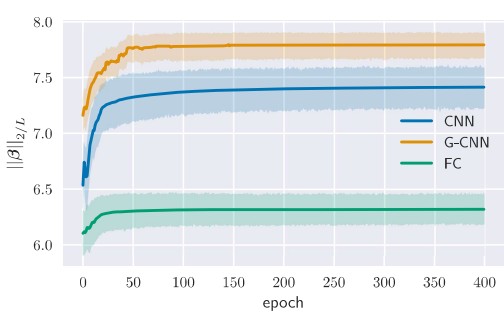

(b) Real space norm of network linearization $\boldsymbol{\beta}$

Figure 1: Training via gradient descent of linear G-CNNs (with linearization $\boldsymbol{\beta}$) implicitly biases towards sparse singular values for Fourier matrix coefficients of $\boldsymbol{\beta}$ (see Theorem 1). Figure 1a traces the $D_8$ Fourier space sparsity over the course of training three architectures in an overparameterized classification task. The G-CNN converges to a Fourier-sparse linearization in contrast with fully-connected and convolutional networks. Figure 1b illustrates the uncertainty principles discussed in Section 6 which show that sparseness in (group) Fourier space necessitates being "dense" in real space, and vice versa. See Appendix D for more visualizations of the implicit bias.

a result, convergence is biased towards sparse solutions in the Fourier regime, as summarized below and illustrated in Figure 1a, as well as further experiments in Section 6.

**Theorem 1** (main result; informal). *Let $\mathrm{NN}_L(\cdot)$ denote an $L$ layer linear group convolutional neural network, in which each hidden layer performs group cross-correlation over the group $G$ with a full-width kernel and the final layer is a fully connected layer. When learning linearly separable data $\{\boldsymbol{x}_i, y_i\}_{i=1}^n$ using the exponential loss function, the network converges in direction to a linear function $\mathrm{NN}_L(\boldsymbol{x}) \propto \boldsymbol{\beta}^T \boldsymbol{x}$, where $\boldsymbol{\beta}$ is given by the following minimization problem:*

$$\min_{\boldsymbol{\beta}} \left\| \widehat{\boldsymbol{\beta}} \right\|_{2/L}^{(S)} \quad s.t. \quad y_i \boldsymbol{x}_i^T \boldsymbol{\beta} \geq 1 \ \ \forall i \in [n] \tag{1}$$

*where $\left\| \widehat{\boldsymbol{\beta}} \right\|_{2/L}^{(S)}$ denotes the $2/L$-Schatten norm of the matrix Fourier transform of $\boldsymbol{\beta}$ (see Definition 4.1) equivalent to*

$$\left\| \widehat{\boldsymbol{\beta}} \right\|_{2/L}^{(S)} = \left[ \sum_{\rho \in \widehat{G}} d_\rho \left( \left\| \widehat{\boldsymbol{\beta}}(\rho) \right\|_{2/L}^{(S)} \right)^{2/L} \right]^{L/2}, \tag{2}$$

*where $\widehat{G}$ is a complete set of unitary irreducible representations of $G$ and $d_\rho$ is the dimension of irreducible representation $\rho$.*

We note that both sparsity (for vectors) and low-rankness (for matrices) are desirable properties for many applications, not least because such predictors are efficient to store and manipulate, thus potentially expanding the scope of application of G-CNNs to areas where sparsity and low-rankness are explicitdesiderata.Connecting our findings to research on uncertainty theorems (Wigderson & Wigderson, 2021) we also show that the implicit regularization towards sparseness (or low rank irreducible representations) in the Fourier regime necessarily implies that solutions in the real regime are "dense," as illustrated in Figure 1b. These results provide a more intuitive and practical perspective into the inductive bias of G-CNNs and the types of functions that they learn.

We proceed as follows. In Section 2, we discuss related works and their relation to our contributions. In Section 3, we define notation. Section 4 provides a basic background in the group theory and Fourier analysis necessary to understand our results. Our main results are stated in Section 5, with the main proof ideas for the abelian (or commutative) and non-abelian (or non-commutative) cases given in Subsection 5.1 and Subsection 5.2, respectively (complete proofs can be found in Appendix A). Section 6 validates our theoretical results with synthetic experiments on a variety of groups and exploratory experiments validating our theory on non-linear networks. Finally, we discuss these results and future questions in Section 7.

## 2 RELATED WORK

Enforcing equivariance and symmetries via parameter sharing schemes was introduced in the group theoretic setting in Cohen & Welling (2016a) and Gens & Domingos (2014). Despite considerable interest in equivariant learning, to the best of our knowledge, no works have explored the implicit regularization of gradient descent on equivariant convolutional neural networks.We show that the tensor formulation of neural networks in Yun et al. (2020) and the proofs in Gunasekar et al. (2018b) encompass G-CNNs for which the underlying group is cyclic and naturally extend to G-CNNs over any commutative group (see Subsection 5.1). However, these works do not cover the case of convolutions with respect to non-commutative groups, such as three-dimensional rotations and permutations, which incidentally include some of the most compelling applications of group equivariance in practice (Zaheer et al., 2017; Anderson et al., 2019; Esteves et al., 2018). As such, articulating the implicit bias in the more general non-abelian case is important for understanding many of the current group equivariant architectures (Zaheer et al., 2017; Kondor et al., 2018; Esteves et al., 2018; Weiler & Cesa, 2019). However, non-abelian convolutions require more structure to theoretically analyze compared to abelian convolutions: the former are merely pointwise multiplications in Fourier space, whereas the latter requires matrix multiplication over irreducible representations which cannot be expressed in the tensor language of Yun et al. (2020). Instead, we use standard optimization tools and comparable convergence assumptions from the work of Gunasekar et al. (2018b) to explicitly characterize the stationary points of convergence for non-abelian G-CNNs.

We also note that our results are consistent with the results of Razin & Cohen (2020) showing that implicit generalization is often captured by measures of complexity which are quasi-norms such as tensor rank. Our results prove that linear G-CNNs are biased towards low rank solutions in the Fourier regime, via regularization of the $2/L$-Schatten norms over Fourier matrix coefficients (also a quasi-norm). Lastly, there is a line of work focusing on understanding the expressivity (Kondor & Trivedi, 2018; Cohen et al., 2019; Yarotsky, 2021) and generalization (Sannai & Imaizumi, 2019; Lyle et al., 2020; Bulusu et al., 2021; Elesedy & Zaidi, 2021) of equivariant networks but not specifically the effects of implicit regularization.

To analyze bounded-width filters, which are more commonly used in practice, a recent work by Jagadeesan et al. (2021) shows that the implicit regularization for an arbitrary filter width $K$ is unlikely to admit a closed-form solution. Separate from calculating the exact form of implicit regularization, there is a rich line of work that details the trade-offs between restricting a function in its real versus Fourier regimes via uncertainty principles (Meshulam, 1992; Wigderson & Wigderson, 2021). While the connection between uncertainty theorems and bounded-width convolutional neural networks has not been thoroughly explored, Caro et al. (2021) and Nicola & Trapasso (2021) highlight the importance of uncertainty principles for understanding the behaviour of modern CNNs.

## 3 NOTATION

Throughout this text, we denote scalars in lowercase script ($a$), vectors in bold lowercase script ($\boldsymbol{a}$), matrices in either bold uppercase script ($\boldsymbol{A}$) or lowercase script hat ($\widehat{a}$) when vectors are transformed into the Fourier regime (see Definition 4.1), and tensors in bold non-italic uppercase script ($\mathbf{A}$). For $f$ a function with range in $\mathbb{C}$, we overload notation slightly and let $\overline{f}$ denote the function with an element-wise complex conjugate applied, i.e. $\overline{f}(x) = \overline{f(x)}$. If $f$ is defined on a group $G$, let $f^-(g) = f(g^{-1})$. For a vector $\boldsymbol{a} \in \mathbb{C}^n$ or a matrix $\boldsymbol{A} \in \mathbb{C}^{m \times n}$, we denote its conjugate transpose as $\boldsymbol{a}^\dagger$ and $\boldsymbol{A}^\dagger$ respectively. We use $\langle \cdot, \cdot \rangle$ to denote the standard vector inner product between two vectors and $\langle \cdot, \cdot \rangle_M$ to denote the inner product between matrices defined as $\langle \boldsymbol{A}, \boldsymbol{B} \rangle_M = \text{tr}(\boldsymbol{A}\boldsymbol{B}^\dagger)$. We use $\| \cdot \|_p$ to denote the vector $p$-norm ($p = 2$ when subscript is hidden) and $\| \cdot \|_p^{(S)}$ to denote the $p$-Schatten norm[1] of a matrix (equivalent to the $p$-vector norm of the singular values of the matrix).

We denote groups by uppercase letters $G$. We denote an irreducible representation (irrep) of a group $G$ by $\rho$ or $\rho_i$ and a complete set of irreps as $\widehat{G}$, so every unitary irrep $\rho$ is equivalent (up to isomorphism) to exactly one element of $\widehat{G}$. The dimension of a given irrep $\rho$ is $d_\rho$.

---

[1]Despite our terminology, $p$-vector and $p$-Schatten norms are technically quasi-norms for $p < 1$.

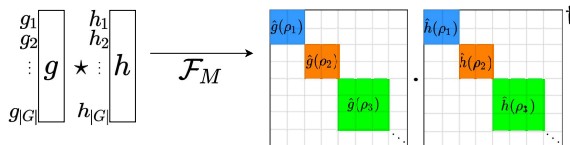

Figure 2: Cross-correlation of two functions over a group is equivalent to matrix multiplication over irreps (shown as blocks of a larger matrix here) in Fourier space.

## 4 BACKGROUND IN GROUP THEORY AND GROUP-EQUIVARIANT CNNs

In this study, we analyze linear G-CNNs in a binary classification setting, where hidden layers perform equivariant operations over a finite group $G$, and networks have no nonlinear activation function after their linear group operations. Inputs $\boldsymbol{x}_i$ are vectors of dimension $|G|$ (*i.e.*, vectorized group functions $\boldsymbol{x} : G \to \mathbb{R}$), and targets $y_i$ are scalars taking the value of either $+1$ or $-1$. Hidden layers in our G-CNNs perform cross-correlation over a group $G$, defined as

$$(\boldsymbol{g} \star \boldsymbol{h})(u) = \sum_{v \in G} \boldsymbol{g}(uv)\boldsymbol{h}(v) \tag{3}$$

Note, that the above is equivariant to the left action of the group, *i.e.*, if $w \in G$ and $\boldsymbol{g}'_w(u) = \boldsymbol{g}(wu)$, then $(\boldsymbol{g}'_w \star \boldsymbol{h})(u) = (\boldsymbol{g} \star \boldsymbol{h})(wu)$. The final layer of our G-CNN is a fully connected layer mapping vectors of length $|G|$ to scalars. We note that this final layer in general will not construct functions that are symmetric to the group operations, as strictly enforcing symmetries in this linear setting will result in trivial outputs (only scalings of the average value of the input). Nonetheless, this model still captures the composed convolutions of practical G-CNNs, and exhibits non-trivial implicit bias.

Analogous to the discrete Fourier transform, there exists a *group* Fourier transform mapping a function into the Fourier basis over the irreps of $G$.

**Definition 4.1** (Group Fourier transform). *Let $f : G \to \mathbb{C}$. Given a fixed ordering of $G$, let $\boldsymbol{e}_u$ be the standard basis vector in $\mathbb{R}^{|G|}$ that is $1$ at the location of $u$ and $0$ elsewhere. Then, $\boldsymbol{f} = \sum_{u \in G} f(u)\boldsymbol{e}_u$ is the vectorized function $\boldsymbol{f}$. Given $\widehat{G}$ a complete set of unitary irreps of $G$, let $\rho \in \widehat{G}$ be a given irrep of dimension $d_\rho$, $\rho : G \longrightarrow \mathrm{GL}\,(d_\rho, \mathbb{C})^2$. The group Fourier transform of $f$, $\widehat{f} : \widehat{G} \to \mathbb{C}$ at a representation $\rho$ is defined as (Terras, 1999)*

$$\widehat{f}(\rho) = \sum_{u \in G} f(u)\rho(u). \tag{4}$$

*By choosing a fixed ordering of $\widehat{G}$, one can similarly construct $\widehat{\boldsymbol{f}}$ as a block-diagonal matrix version of $\widehat{f}$ (as in Figure 2). We define $\mathcal{F}_M$ to be the* matrix *Fourier transform that takes $\boldsymbol{f}$ to $\widehat{\boldsymbol{f}}$:*

$$\widehat{\boldsymbol{f}} = \mathcal{F}_M \boldsymbol{f} = \bigoplus_{\rho \in \widehat{G}} \widehat{f}(\rho)^{\oplus d_\rho} \ \in \mathrm{GL}\,(|G|, \mathbb{C})\,. \tag{5}$$

*$\widehat{\boldsymbol{f}}$ or $\mathcal{F}_M \boldsymbol{f}$ are shortened notation for the complete Fourier transform. Furthermore, by vectorizing the matrix $\widehat{\boldsymbol{f}}$, there is a* unitary *matrix $\mathcal{F}$ taking $\boldsymbol{f}$ to $\widehat{\boldsymbol{f}}$, analogous to the standard discrete Fourier matrix. We use the following explicit construction of $\mathcal{F}$: denoting $\boldsymbol{e}_{[\rho,i,j]}$ as the column-major vectorized basis for element $\rho_{ij}$ in the group Fourier transform, then we can form the matrix*

$$\mathcal{F} = \sum_{u \in G} \sum_{\rho \in \widehat{G}} \frac{\sqrt{d_\rho}}{\sqrt{|G|}} \sum_{i,j=1}^{d_\rho} \rho(u)_{ij} \boldsymbol{e}_{[\rho,i,j]} \boldsymbol{e}_u^T. \tag{6}$$

*Intuitively, for each group element $g$, the matrix $\mathcal{F}$ contains all the irrep images $\rho(g)$ 'flattened' into a single column. See Appendix B for further exposition.*

Convolution and cross-correlation are equivalent, up to scaling, to matrix operations after Fourier transformation. For example, in the case of cross-correlation (Equation 3), $\widehat{(g \star h)}(\rho) = \widehat{g}(\rho)\widehat{h}(\rho)^\dagger$. This simple fact, illustrated in Figure 2, is the basis for the proofs of our implicit bias results.

---

[2]Note that for an abelian group, $d_\rho = 1 \ \forall \rho$. For standard Fourier analysis over the cyclic group, each $\rho$ is a complex sinusoid at some frequency.

## 5 MAIN RESULTS

We consider linear group-convolutional networks for classification analogous to those of Yun et al. (2020) and Gunasekar et al. (2018b). A linear G-CNN is composed of several group cross-correlation layers followed by a fully connected layer. The input is formalized as a function on the group (according to some pre-defined ordering of elements), $\boldsymbol{x} : G \to \mathbb{R}$, and the output is a scalar. Explicitly, let $G$ be a finite group with $\boldsymbol{x}, \boldsymbol{w}_1, \ldots, \boldsymbol{w}_L$ real-valued functions from $G$ to $\mathbb{R}$. The network output is $\mathrm{NN}(\boldsymbol{x}) = \langle \boldsymbol{x} \star \boldsymbol{w}_1 \star \cdots \star \boldsymbol{w}_{L-1}, \boldsymbol{w}_L \rangle \triangleq \langle \boldsymbol{x}, \boldsymbol{\beta} \rangle$. Let $\boldsymbol{W} = [\boldsymbol{w}_1 \ \ldots \ \boldsymbol{w}_L]$ be the concatenation of all network parameters, and $\boldsymbol{\beta} = \mathcal{P}(\boldsymbol{W}) = \boldsymbol{w}_L \star \boldsymbol{w}_{L-1}^- \star \cdots \star \boldsymbol{w}_1^-$ be the "end-to-end" linear predictor consisting of composed cross-correlation operations. One can check (as we do in Lemma A.1) that $\widehat{\boldsymbol{\beta}} \triangleq \mathcal{F}_M \boldsymbol{\beta} = \widehat{\boldsymbol{w}_L} \ldots \widehat{\boldsymbol{w}_1}$. Networks are trained via gradient descent over the exponential loss function on linearly separable data $\{\boldsymbol{x}_i, y_i\}_{i=1}^N$. Iterates take the form

$$\boldsymbol{W}_\ell^{(t+1)} = \boldsymbol{W}_\ell^{(t)} - \eta_t \nabla_{\boldsymbol{W}_\ell} \mathcal{L}(\mathcal{P}(\boldsymbol{W})) \tag{7}$$

where $\ell(\langle \boldsymbol{x}, \boldsymbol{\beta} \rangle, y) = \exp(-\langle \boldsymbol{x}, \boldsymbol{\beta} \rangle \cdot y)$ and $\mathcal{L}(\boldsymbol{\beta}) = \sum_{i=1}^N \ell(\langle \boldsymbol{x}_i, \boldsymbol{\beta} \rangle, y_i)$.

### 5.1 ABELIAN

Similar to ordinary (cyclic) convolution[3], the commutative property of abelian groups implies that convolutions in real space are equivalently pointwise multiplication of irreps in Fourier space since all irreps are one-dimensional for commutative groups ($d_\rho = 1 \forall \rho$). To start, recall the key definition of Yun et al. (2020), determining which network architectures fall within the purview of their results:

**Proposition 5.1** (paraphrased from Yun et al. (2020)). *Let $\mathbf{M}(\boldsymbol{x})$ be a map from data $\boldsymbol{x} \in \mathbb{R}^d$ to a data tensor $\mathbf{M}(\boldsymbol{x}) \in \mathbb{R}^{k_1 \times k_2 \times \cdots \times k_L}$. The input into an $L$-layer tensorized neural network can be written as an orthogonally decomposable data tensor if there exists a full column rank matrix $\boldsymbol{S} \in \mathbb{C}^{m \times d}$ and semi-unitary matrices $\boldsymbol{U}_1, \ldots, \boldsymbol{U}_L \in \mathbb{C}^{k_\ell \times m}$ where $d \leq m \leq \min_\ell k_\ell$ such that:*

$$\mathbf{M}(\boldsymbol{x}) = \sum_{j=1}^m [\boldsymbol{S}\boldsymbol{x}]_j \left([\boldsymbol{U}_1]_{\cdot,j} \otimes [\boldsymbol{U}_2]_{\cdot,j} \otimes \cdots \otimes [\boldsymbol{U}_L]_{\cdot,j}\right), \tag{8}$$

*such that the network output is the tensor multiplication between $\mathbf{M}(x)$ and each layer's parameters:*

$$\mathrm{NN}(\boldsymbol{x}; \Theta) = \mathbf{M}(\boldsymbol{x}) \cdot (\boldsymbol{w}_1, \ldots, \boldsymbol{w}_L) = \sum_{i_1=1}^d \cdots \sum_{i_L=1}^d \mathbf{M}(x)_{i_1 \ldots i_L} (\boldsymbol{w}_1)_{i_1} \cdot \cdots \cdot (\boldsymbol{w}_L)_{i_L}.$$

Indeed, a linear G-CNN over an abelian group can be expressed in a way that satisfies Proposition A.3 for an appropriate choice of $\boldsymbol{S}$ and $\boldsymbol{U}_1, \ldots, \boldsymbol{U}_L$, as stated in the following proposition.

**Proposition 5.2.** *Let $\mathbf{M}(\boldsymbol{x})$ be an orthogonally decomposable data tensor with associated matrices $\boldsymbol{S}, \boldsymbol{U}_1, \ldots, \boldsymbol{U}_L$ as in Proposition 5.1. Given a finite abelian group $G$, let $d = m = k_\ell = |G|$ and $\mathcal{F} \in \mathbb{C}^{d \times d}$ be the group Fourier transform of $G$ (see Definition 4.1). With $\boldsymbol{S} = d^{\frac{L-1}{2}} \mathcal{F}$, unitary matrices $\boldsymbol{U}_\ell = \mathcal{F}^{-1} = \mathcal{F}^\dagger$, and the data tensor $\mathbf{M}(\boldsymbol{x})$ defined correspondingly, the output of a G-CNN with real-valued filters $\boldsymbol{w}_1, \ldots, \boldsymbol{w}_L$ is a tensor operation:*

$$\mathbf{M}(\boldsymbol{x}) \cdot (\boldsymbol{w}_1, \ldots \boldsymbol{w}_L) = \langle \boldsymbol{x} \star \boldsymbol{w}_1 \star \cdots \star \boldsymbol{w}_{L-1}, \boldsymbol{w}_L \rangle$$

The proof is deferred to Appendix A. Fundamentally, the result requires not only that $\mathcal{F}$ is unitary, which holds for all finite groups, but also that cross-correlation is pointwise multiplication (up to a conjugate transpose) in Fourier space, i.e. $\mathcal{F}(\boldsymbol{x} \star \boldsymbol{w}) = (\mathcal{F}\boldsymbol{x}) \odot (\mathcal{F}\boldsymbol{w})^\dagger$. This property only holds for commutative groups, as matrix multiplication is pointwise multiplication only for matrices of dimension $d_\rho = 1$. Given Proposition 5.2, we apply the implicit bias statement of Yun et al. (2020).

**Theorem 5.3** (Implicit regularization of linear G-CNNs for $G$ an abelian group). *Suppose there exists $\lambda > 0$ such that the initial directions $\bar{\boldsymbol{w}}_1, \ldots, \bar{\boldsymbol{w}}_L$ of the network parameters satisfy*

---

[3]In fact, we include a canonical result in the appendix, Theorem A.4, demonstrating that *all* finite abelian groups are direct products of cyclic groups, i.e. *multidimensional* translational symmetries.

$\left|[\mathcal{F}\bar{\boldsymbol{w}}_\ell]_j\right|^2 - \left|[\mathcal{F}\bar{\boldsymbol{w}}_L]_j\right|^2 \geq \lambda$ *for all* $\ell \in [L-1]$ *and* $j \in [m]$, *i.e. if the Fourier transform magnitudes of the initial directions look sufficiently different pointwise (which occurs with high probability for e.g. a Gaussian random initialization). Then,* $\boldsymbol{\beta} = \mathcal{P}([\boldsymbol{w}_1, \ldots, \boldsymbol{w}_L])$ *converges in a direction that aligns with a stationary point* $\boldsymbol{z}_\infty$ *of the following optimization program:*

$$\min_{\boldsymbol{z} \in \mathbb{C}^m} \|\mathcal{F}\boldsymbol{z}\|_{2/L} \quad s.t. \quad y_i\langle \boldsymbol{x}_i, \boldsymbol{z}\rangle \geq 1, \forall i \in [n] \tag{9}$$

As noted in Theorem A.4 of the Appendix, all finite abelian groups can be expressed as a direct product of cyclic groups. In contrast, many groups (rotations, subgroups of permutations, etc.) with much richer structure are non-commutative, and we now turn our attention to the non-abelian case.

## 5.2 NON-ABELIAN

In Fourier space, non-abelian convolution consists of matrix multiplication over irreps, and do *not* fit the pointwise multiplication structure of Proposition 5.2. We instead build upon the results of Gunasekar et al. (2018b), and directly analyze the stationary points of the proposed optimization program to prove the following:

**Theorem 5.4** (Non-abelian; see also Theorem A.5). *Consider a classification task with ground-truth linear predictor* $\boldsymbol{\beta}$, *trained via a linear G-CNN architecture with* $L > 2$ *layers under the exponential loss. For almost all* $\boldsymbol{\beta}$-*separable datasets* $\{\boldsymbol{x}_i, y_i\}_{i=1}^n$, *any bounded sequence of step sizes* $\eta_t$, *and almost all initializations: if the loss converges to 0, the gradients converge in direction, and the iterates themselves all converge in direction to a classifier with positive margin, then the resultant predictor is a scaling of a first order stationary point of the optimization problem:*

$$\min_{\boldsymbol{\beta}} \|\widehat{\boldsymbol{\beta}}\|_{2/L}^{(S)} \quad s.t. \quad \forall n, y_n\langle \boldsymbol{x}_n, \boldsymbol{\beta}\rangle \geq 1. \tag{10}$$

To prove the above statement, we show that linear G-CNNs converge to stationary points of the KKT conditions above, which is also the high-level method of Gunasekar et al. (2018b). However, our proof differs in several key ways from that of Gunasekar et al. (2018b). First, we redefine operations of the G-CNN as a series of inner products and cross-correlations over the space of the matrix Fourier transform of Definition 4.1. Second, in this Fourier space, we analyze the subdifferential of the Schatten norms to ultimately show that the KKT conditions of Equation 10 are satisfied. In contrast, Gunasekar et al. (2018b) analyze the subdifferential of a different objective, the ordinary $2/L$-vector norm. The fact that the irreps of a group are only unique up to isomorphism (*e.g.,* conjugation by a unitary matrix) aids in identifying this Schatten norm as the correct regularizer, since Schatten norms are invariant to unitary matrix conjugation. These features are specific to the non-abelian case. More specifically, the proof of this result follows the outline below:

1. First, by applying a general result of Gunasekar et al. (2018b), Theorem A.6, we characterize the implicit regularization in the full space of parameters, $\boldsymbol{W}$ (in contrast to the end-to-end linear predictor $\boldsymbol{\beta}$), as a (scaled) stationary point $\boldsymbol{W}^\infty$ of the following optimization problem in $\boldsymbol{W}$:

$$\min_{\boldsymbol{W} \in \mathbb{R}^P} \|\boldsymbol{W}\|_2^2 \quad s.t. \quad \forall n, y_n\langle \boldsymbol{x}_n, \mathcal{P}(\boldsymbol{W})\rangle \geq 1 \tag{11}$$

2. Separately, we define a *distinct* optimization problem, Equation 10, in $\boldsymbol{\beta}$ to show that stationary points of Equation 11 are a subset of those of Equation 10, up to scaling.

3. The *necessary* KKT conditions for Equation 11 characterize its stationary points:

$$\exists\{\alpha_n : \alpha_n \geq 0\}_{n=1}^N \text{ s.t. } \alpha_n = 0 \text{ if } y_n\langle \boldsymbol{x}_n, \mathcal{P}(\boldsymbol{W}^\infty)\rangle > 1$$

$$\boldsymbol{w}_i^\infty = \nabla_{\boldsymbol{w}_i}\mathcal{P}(\boldsymbol{W}^\infty)\left[\sum_n \alpha_n y_n \boldsymbol{x}_n\right] = \nabla_{\boldsymbol{w}_i}\left\langle\mathcal{P}(\boldsymbol{W}^\infty), \sum_n \alpha_n y_n \boldsymbol{x}_n\right\rangle \tag{12}$$

From here, we show that the *sufficient* KKT conditions for Equation 10 are also satisfied by the corresponding end-to-end predictor. In particular, we calculate the set of subgradients[4] $\partial^o\|\widehat{\boldsymbol{\beta}}\|_p^{(S)}$ for $p = \frac{2}{L} < 1$, and then use Equation 12 to derive recurrences demonstrating that a positive scaling of $\sum_n \alpha_n y_n \widehat{\boldsymbol{x}}_n$ is a member of this set.

---

[4]$\partial^o$ is the local subgradient of Clarke (1975): $\partial^o f(\boldsymbol{\beta}) = \text{conv}\{\lim_{i\to\infty} \nabla f(\boldsymbol{\beta} + \boldsymbol{h}_i) : \boldsymbol{h}_i \to 0\}$

**Remark.** *For abelian groups where all irreps are one-dimensional, $\widehat{\beta}$ in Theorem 5.4 is a diagonal matrix. Thus, the $p$-Schatten norm coincides with the $p$-vector norm of the diagonal entries, recovering results in Subsection 5.2. However, Theorem 5.4 requires stronger convergence assumptions.*

**Infinite dimensional groups:** Theorem 5.4 applies to all *finite* groups, but G-CNNs have extensive applications for *infinite* groups, where outputs of convolutions are infinite-dimensional. Here, it is common to assume sparsity in the Fourier coefficients and "band-limit" filters over a set of low-frequency irreps (under some natural group-specific ordering) that form a finite dimensional linear subspace (we denote the representation of a function in this band-limited Fourier space as $\widehat{\underline{w}}$.) Here, G-CNNs with band-limited filters take precisely the form of the finite G-CNNs from Theorem 5.4. Thus, slightly modifications yield the following for infinite groups (see Appendix A.2.1 for details).

**Corollary 5.5** (Infinite-dimensional groups with band-limited functions; see also Theorem A.11). *Let $G$ be a compact Lie group with irreps $\widehat{G}$, and let $B \subset \widehat{G}$ with $|B| < \infty$.[5] Proceed fully in Fourier space, in the subspace corresponding to $B$: consider a linearly separable classification task with ground-truth linear predictor $\underline{\widehat{\beta}}$ and inputs $\widehat{\underline{x}}$, both real-valued and supported only on irreps in $B$, and proceed by gradient descent on the band-limited Fourier-space filters. Under near-identical conditions as Theorem 5.4, the resultant predictor is a scaling of a first order stationary point of:*

$$\min_{\underline{\widehat{\beta}}} \left\| \underline{\widehat{\beta}} \right\|_{2/L}^{(S)} \quad s.t. \quad \forall n, y_n \left\langle \widehat{\underline{x}}_n, \underline{\widehat{\beta}} \right\rangle \geq 1. \tag{13}$$

## 6 EXPERIMENTS

Here, we first experimentally confirm our theory in a simple setting illustrating the effects of implicit regularization. Then, we relax the crucial assumption of linearity in our setup to empirically show that our results may hold locally even in nonlinear settings. Note that the results for nonlinear networks in Subsection 6.2 are *only* empirical in nature, and Theorems 5.3 and A.5 do not necessarily hold in the more general nonlinear setting. For all binary classification tasks, we use three-layer networks with inputs and convolution weights in $\mathbb{R}^{|G|}$. Since we are interested in the resulting implicit bias alone, we only analyze loss on data in the training set. A complete description of our experimental setup can be found in Appendix E.

Throughout this section, we plot norms in the Fourier and real regimes side-by-side to highlight unavoidable trade-offs in implicit regularization between the two conjugate regimes. These trade-offs, which have a rich history of study in physics and group theory, are commonly termed uncertainty principles (Wigderson & Wigderson, 2021). One especially relevant uncertainty theorem states that sparseness in the real or Fourier regime necessarily implies dense support in the conjugate regime.

**Theorem 6.1** (Meshulam uncertainty theorem (Meshulam, 1992)). *Given a finite group $G$ and $f : G \to \mathbb{C}$, let $\widehat{G}$ be the set of irreps of $G$ and $\boldsymbol{f}$ be the vectorized function (see Definition 4.1). Then*

$$| \operatorname{supp}(\boldsymbol{f})| \, \operatorname{rank}(\widehat{\boldsymbol{f}}) = | \operatorname{supp}(\boldsymbol{f})| \sum_{\rho \in \widehat{G}} d_\rho \operatorname{rank}\left(\widehat{\boldsymbol{f}}(\rho)\right) \geq |G| \tag{14}$$

Other related uncertainty principles are detailed in Appendix C.

### 6.1 EMPIRICAL CONFIRMATION OF THEORY

Here, we trace the regularization through (training) epochs via analysis over three groups which are all trained to classify data with $\pm 1$ labels:

- $D_8$ (Figure 1): The dihedral group $D_8$ is a simple non-abelian group that captures the geometry of a square[6]. Inputs are vectors with elements drawn i.i.d. from $N(0, 1)$.
- $(C_5 \times C_5) \rtimes Q_8$ (Figure 3): A non-abelian group that has irreducible representations of up to dimension 8 displays implicit regularization over a more elaborate group structure. Inputs are vectors with elements drawn i.i.d. from the standard Normal distribution.

---

[5] For example, $G = SO(3)$ and $B$ indexes all Wigner d-matrices with $|\ell| \leq L$ (Kondor et al., 2018).

[6] We use the convention that $D_n$ is the dihedral group of order $n$.

- $(C_{28} \times C_{28}) \times D_8$ (Figure 4): A non-abelian group which acts on images (the digits 1 and 5) from the MNIST dataset.

In the three settings above, we compare the behaviors of G-CNN, traditional CNN[7], and fully-connected (FC) network architectures with similar instantiations. We plot the the real space vector norm and Fourier space Schatten norm of the network linearization over training epochs. All models perfectly fit the data in this overparameterized setting, and convergence to a given regularized solution corresponds to convergence in the loss to zero.

Consistent with theory, G-CNN architectures shown in Figures 1, 3, and 4 have the smallest Fourier space Schatten norms among the architectures. FC networks exhibit no group Fourier regime regularization while standard CNNs exhibit some regularization since they share some irreducible representations with the G-CNN. The differing behaviors of CNNs and FC networks show that implicit regularization is a consequence of the choice of architecture and not inherent to the task itself.

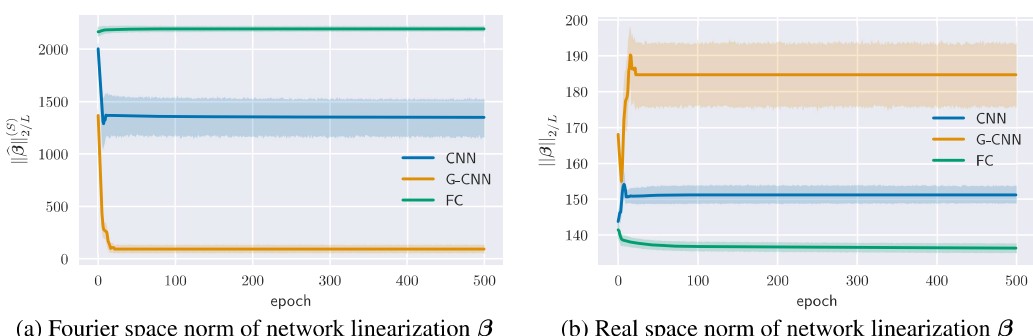

(a) Fourier space norm of network linearization $\beta$     (b) Real space norm of network linearization $\beta$

Figure 3: Norms of the linearizations of three different linear architectures for the non-abelian group $G = (C_5 \times C_5) \rtimes Q_8$ trained on a binary classification task with 10 isotropic Gaussian data points.

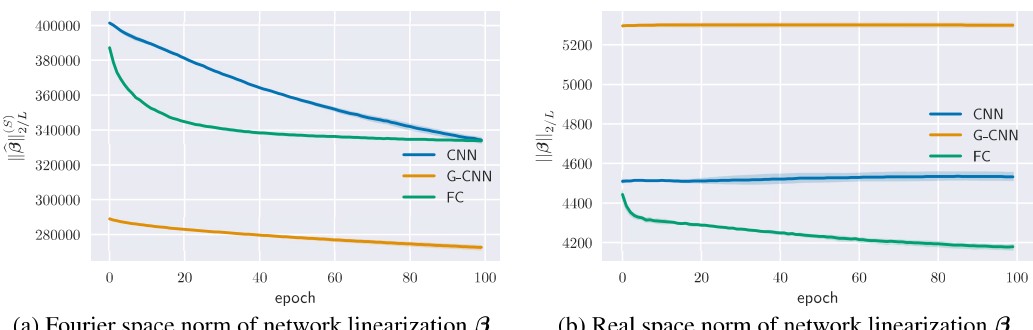

(a) Fourier space norm of network linearization $\beta$     (b) Real space norm of network linearization $\beta$

Figure 4: Norms of the linearizations of three different linear architectures for the non-abelian group $G = (C_{28} \times C_{28}) \times D_8$ trained using the digits 1 and 5 from the MNIST dataset.

## 6.2 ASSESSMENT OF THEORY ON NONLINEAR NETWORKS

Here we introduce rectified linear unit (ReLU) nonlinearities between hidden layers to analyze implicit bias in the presence of nonlinearity. Our theoretical results do not necessarily hold in this case, so we are *exploring, rather than confirming,* their validity for nonlinear networks. Accordingly, given a G-CNN with ReLU activations, we wish to calculate the Schatten norm of the Fourier matrix coefficients of the network linearization $\beta$. However, networks can no longer be collapsed into linear functions due to the nonlinearities. Instead, we construct local linear approximations at each data point via a first-order Taylor expansion, calculate the norms of interest according to this linearization, and average the results across the dataset to get a single aggregate value. We evaluate the implicit bias of an invariant ReLU G-CNN (with final pooling layer) with respect to translations,

---

[7]We generically use "CNN" to refer to a G-CNN over the cyclic group of size equal to the size of the input.

rotations, and flips on MNIST digits, and a nonlinear G-CNN (with linear final layer) on the dihedral group $D_{60}$ with synthetic data. Note that we use linear approximations only to analyze implicit regularization, and not to further interpret the outputs of the locally linear neural network, as such analysis can give rise to misleading or fragile feature importance maps (Ghorbani et al., 2019).

Remarkably, as shown in Figures 5a and 5b, our results remain valid in this nonlinear setting. While this does not guarantee that our implicit bias characterization will hold in more general settings, it is an encouraging sign that our theoretical results handle the violation of the linearity assumption, at least numerically. Additional figures detailing the real-space behaviour are provided in Appendix E.

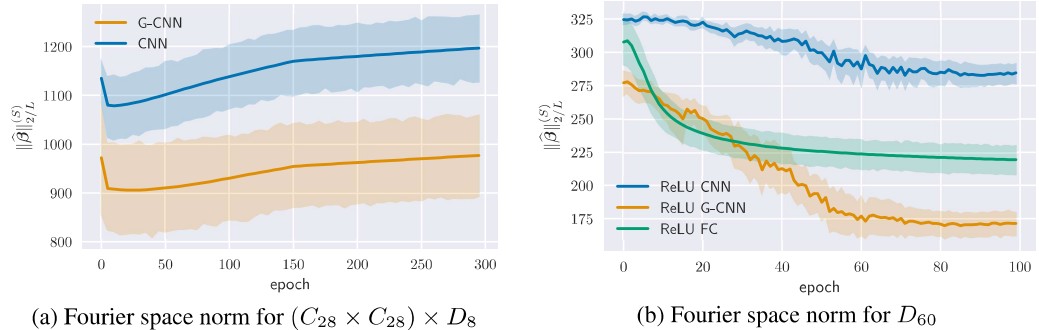

(a) Fourier space norm for $(C_{28} \times C_{28}) \times D_8$      (b) Fourier space norm for $D_{60}$

Figure 5: Group Fourier norms for *nonlinear* architectures with ReLU activations show that nonlinear G-CNNs seem to implicitly regularize locally. Both figures track the mean norm of the per-sample local linearizations of each network. The networks in Figure 5a use a pooling layer after the convolutional layers to maintain invariance. Figure 5a evaluates a binary classification task using the MNIST digits 0 and 5. Figure 5b is obtained on networks with a final linear layer, and evaluates a binary classification task with 10 isotropic Gaussian data points.

## 7 DISCUSSION

In this work, we have shown that $L$-layer linear G-CNNs with full width kernels are biased towards sparse solutions in the Fourier regime regularized by the $2/L$-Schatten norm over Fourier matrix coefficients. Our analysis applies to linear G-CNNs, over finite groups (or infinite groups with band-limited inputs), trained in the task of binary classification. In advancing our results on implicit regularization, we highlight some limitations of our work and important future directions:

- **Nonlinearities:** Adding nonlinearities to the G-CNNs studied here expands the space of functions which the G-CNNs can express, but implicit regularization in this nonlinear setting may be challenging to characterize as G-CNNs are no longer linear predictors. Local analysis may be possible in special cases, *e.g.,* in the infinite width limit (Lee et al., 2017).

- **Bounded width kernels:** Our results apply to full-width kernels with support on the whole space of group operations. Expanding results to bounded-width kernels, *i.e.,* those with sparse support, is an obvious future direction, though prior work has indicated that closed form solutions may not exist for these more special cases (Jagadeesan et al., 2021).

- **Different loss function and learning settings:** The loss function studied here is the exponential loss function used in binary classification tasks. Beyond binary classification, it is an open question how the implicit regularization changes for classification over more than two classes, not only for G-CNNs but for other re-parametrizations like CNNs and fully-connected networks as well (Gunasekar et al., 2018b).

Although concise implicit regularization measures are challenging to analyze for realistic, nonlinear architectures, linear networks provide an instructive case study with precise analytic results. In this work, by proving that linear group equivariant CNNs trained with gradient descent are regularized towards low-rank matrices in Fourier space, we hope to advance the broader agenda of understanding generalization, and in particular how and why networks with diverse architectures — particularly those with built-in symmetries — learn.

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
