# OpenReview forum: "Implicit Bias of Linear Equivariant Networks"
_ICLR.cc/2022/Conference — ICLR 2022 Submitted_

### Official Review · Reviewer_HWqt · 2021-10-31

**Correctness:** 3
**Technical Novelty And Significance:** 3
**Empirical Novelty And Significance:** 2
**Recommendation:** 6
**Confidence:** 3

**Main Review:**

The authors show for the models under consideration an interesting and novel result: the models are implicitly biased not by the real space norm of the linear predictor, but by the Schatten norm of the Fourier transform of the predictor. For the Abelian case, this required an incremental generalization of prior work, but for the non-Abelian case required an interesting proof method. The paper is very clearly written and was a pleasure to read.

My main problem with this work is that it considers a model class that I’ve never seen used (besides the lack of non-linearities): it uses group equivariant CNNs and then takes an inner product with a G-feature weight, resulting in a non-equivariant linear predictor. If the authors used invariant outputs, the linear invariant predictors would just be averaging, and the analysis wouldn’t work. As G-CNNs are chosen for their symmetry properties, studying them in a non-equivariant context makes little sense to me. Whether the implicit bias found in this paper generalizes to equivariant non-linear G-CNN networks has not been convincingly shown.

For me to recommend acceptance of this paper, the authors should make a more convincing point why the results for this non-equivariant model class are applicable to equivariant G-CNN networks people actually use (besides the fact that the analysis applies to linear networks).

Other weaknesses:
* The non-Abelian case states two assumptions (gradients converge in direction, iterates converge in direction to a classifier with positive margin), but these are not clearly defined, nor is it stated when these are satisfied. The paper would be improved if there would be some discussion of these assumptions.
* The key point of the paper seems to be that Fourier Schatten norm, not spatial norm is the implicit bias of the learned networks. The paper would be stronger if it would give some more intuition about this point. Perhaps visualize some of the learned weights in both Fourier and spatial domain?

**Summary Of The Paper:**

Group Equivariant CNNs (G-CNNs) generalize CNNs from the translation group to arbitrary groups. It has been shown that overparameterized/undetermined linear transformations and linear CNNs learned by gradient descent are implicitly biased, in that they find zero train loss solutions that minimize some norm on the predictor. This paper generalizes these results from linear CNNs to linear G-CNNs. The linear G-CNNs are constructed into a non-equivariant linear transformation, by doing L G-CNN layers, followed by a non-equivariant inner product with a G-feature. The authors show that the implicit bias is to minimize the p-norm of the singular values (the Schatten norm) of the Fourier transform of the resulting linear transformation.
In the case where G is Abelian, by a tensor factorization of the linear G-CNN, the authors can straightforwardly apply a result from prior work on the implicit bias of such tensor factorizations to prove their result.
In the case where G is not Abelian, the authors prove their results in three steps. (1) They use prior work on gradient descent on linear predictors that are polynomials of the parameters, which their linear G-CNN falls under, to show that the stationary points minimize norm the joint norm on all weights. (2) They define a linear regression optimization that minimizes the Schatten norm of the linear predictor. (3) They show that all optima of the polynomial linear predictor are also optima of the Schatten-normed linear regression. To do so, they find the subdifferential of the p-Schatten norm and show that the polynomial optimum is in the subdifferential.

The authors confirm their theory in toy experiments in the linear case and show that in the non-linear case, where the Schatten norm is computed by local linearization, that similar results may hold in the non-linear case.

**Summary Of The Review:**

While the authors use novel methods to derive an interesting implicit bias of the model class under consideration, this class, non-equivariant predictor based on equivariant neural network layers, is not what is generally used in practice. For the paper to be recommendable for acceptance, the authors should make a stronger argument why their results also apply to models actually used. If they do so convincingly, I'd increase my score.



Updated score to 6.

---

> ### Author Response · Authors · 2021-11-23
> **Response to Reviewer HWqt**
>
> We thank the author for their valuable feedback and directing our attention to analyzing models that are used in practice.
>
> This reviewer rightly points out that the linear networks we studied are a simplification of the models people use in practice. From a theoretical standpoint, our idealization is not fully invariant; as the reviewer notes, the only truly invariant linear model simply averages the input data, which is not an interesting case to analyze from an implicit bias perspective. However, this formalism does allow us to analyze the properties of networks with several “stacked” invariant layers, even though the full network is not end-to-end invariant. We argue that this case is still valuable: after all, such networks intuitively will learn high-level invariant features up to the last layer. Our ultimate finding that the network is biased towards the 2/L-Schatten norm of the Fourier regime shows that such a network outputs functions which are still strongly connected to the properties of the group. For example, the 2/L-Schatten norm is a norm that is invariant to the choice of (non-unique) irreducible representations of the group.
>
> In addition, we understand that our numerical simulations could have more closely analyzed the more practical setting to observe if such implicit bias is observed at least locally. In response, we have performed additional numerical analysis on more practical networks to test this and find that such implicit bias is still observed in this setting, in relation to other network architectures (fully-connected and vanilla convolutional). We added experiments confirming our theory on linear G-CNNs trained on the MNIST classification dataset with respect to the group of translations, 90 degree rotations, and flips, and on nonlinear G-CNNS (with respect to their local linear approximations). Moreover, we repeated this experiment with a nonlinear G-CNN with ReLU activations, and found that our theory was still predictive of the local linearization: the G-CNN converged to a predictor with lower Schatten norm than an ordinary CNN. We respond further to specific comments below.
>
> **The non-Abelian case states two assumptions (gradients converge in direction, iterates converge in direction to a classifier with positive margin), but these are not clearly defined, nor is it stated when these are satisfied. The paper would be improved if there would be some discussion of these assumptions.**
>
> We included full details of our assumptions in the appendix (see Theorem A.5), due to space constraints in the main body of the paper, and have further clarified them there in response to this comment.
>
> **The key point of the paper seems to be that Fourier Schatten norm, not spatial norm is the implicit bias of the learned networks. The paper would be stronger if it would give some more intuition about this point. Perhaps visualize some of the learned weights in both Fourier and spatial domain?**
>
> We agree that the form of the implicit bias may not be directly evident and further intuition will aid the reader. In light of this, we have included in Appendix D, clear visualizations of the bias pre- and post-training for each of the networks we study. The sparseness and low-rank bias of the G-CNN in the Fourier regime is readily apparent in this figure. Furthermore, one can observe the lack of implicit bias in fully connected architectures as well as the uncertainty principle trade-offs for the G-CNN.

---

> > ### Comment · Reviewer_HWqt · 2021-11-29
> > **Thanks for the revision and response**
> >
> > Thank you for the revision and the response. I'll increase my score to a 6, because of the GCNN experiment and the weight visualisations. I remain to have doubts about the applicability of the theory, as the linear regimes necessitates learning non-equivariantly to make the theory work, which is not what is done in practice.

---

### Official Review · Reviewer_nEDA · 2021-11-02

**Correctness:** 4
**Technical Novelty And Significance:** 3
**Empirical Novelty And Significance:** 2
**Recommendation:** 6
**Confidence:** 2

**Main Review:**

Strengths:
As far as I can assess, the paper is technically sound, novel and timely.
The paper is formal and precise.
The appendix is quite nice actually and is rather complete in mathematical background.
The paper is reproducible and has code provided.

Weaknesses:
My main criticism is that the paper is very theoretical and feels a bit like a nice math exercise to prove something, however, without being clear why the problem is addressed in the first place (apart from the fact that no one did it yet). To me, as someone working with G-CNNs and having a good understanding of (irreducible) representation theory, but not in analysis, it is unclear what I should take home from the paper.
I.e., why should we expect different results for non-Abelian compared to Abelian groups in the first place?
How should I interpret the bias towards dense solutions as a regularization effect?
What are the implications?

Comments:
It would be helpful to have explicit statements about the main contributions/novelty of the paper.

The experiments in the figure make sense to me in the comparison G-CNN vs FC. Both map G-feature maps to scalars and the latter shows that the bias takes place due to architecture and not the data. However, I do not understand the comparison CNN to G-CNN. If the feature maps are functions on G, is a CNN not automatically a G-CNN? Then, what is the difference between the two?

Some essential details are missing regarding the data. I cannot find it in the appendix either. The closest I get to a description is in caption of Figure 3: “trained … with six isotropic Gaussian data points”. What is an isotropic Gaussian data point?

In theorem 5.3 third line: is this a typo? (w.r.t. indexing to l should be l+1?, it says something like Fw_l-Fw_l \geq \lambda)


**Summary Of The Paper:**

The paper describes a theoretical analysis of the implicit bias in G-CNNs. The results show that for linear G-CNNs trained on linearly separable data using GD converge to sparse solutions in the Fourier domain (equivalently dense solutions in the real domain). The theory is theoretically confirmed and shows that the implicit bias also occurs to some extend for non-linear G-CNNs.

**Summary Of The Review:**

Although I am comfortable with group theory and basic harmonic analysis on groups through irreps, I find it hard to follow the derivations and make sense out of the statements. My impression is that the paper is too mathematical for ICLR audience, though I can really only speak for myself. The paper could improve a lot from more layman's explanations throughout the paper. For example, it already starts early in the paper with talk about Schatten norms, where to me it is not immediately obvious why these are considered.

I understand that it is hard to please every reader, but I believe it is possible to convey the main messages better to a broader audience. Currently it seems overly tuned towards experts in analysis (of implicit biases).

Overall, I do see value in publishing such work as the paper seems otherwise sound.

---

> ### Author Response · Authors · 2021-11-23
> **Response to Reviewer nEDA (1/2)**
>
> We thank the reviewer for their thoughtful comments and critiques. Since other reviewers had asked us to provide more intuition behind our results, we have provided more visualizations of our results and have analyzed the bias imparted by a broader set of more practical equivariant neural networks. Though we will discuss some of these in our response below, we would point the reviewer as well to the open-ended response directed at all reviewers.
>
> Below, we proceed to respond to individual points of criticism.
>
> **it is unclear what I should take home from the paper. I.e., why should we expect different results for non-Abelian compared to Abelian groups in the first place? How should I interpret the bias towards dense solutions as a regularization effect? What are the implications?**
>
> Since linear equivariant networks are capable of outputting any linear function, and many linear functions perfectly fit linearly separable training data in the overparametrized regime, it is unclear a priori what overparametrized G-CNNs converge to. Our results settle this question by showing that solutions converge to a linear separator with minimal $2/L$-Fourier Schatten norm. This norm has a group theoretic “structure,” since it’s invariant to conjugations of the group irreps.
>
> For an intuitive understanding of the abelian vs non-abelian case, we offer two perspectives. First, by the fundamental theorem of abelian groups (see Appendix A.1.1), any abelian group is equivalent to a direct product of cyclic groups. In other words, any abelian group looks like cyclic translations in some number of dimensions. In contrast, non-abelian groups are a far richer family of transformations and cannot be so cleanly characterized. Second, in Fourier space, convolution with respect to an abelian group looks like scalar multiplication. However, convolution with respect to a non-abelian group looks like a series of stacked matrix multiplications, which leads to more complication.
>
> Stepping outside the idealistic setting of linear G-CNNs, our results raise the conjecture that a wide class of equivariant networks are biased towards sparsity and low-rankness in the Fourier regime. Our numerical simulations provide evidence towards this conjecture, and we would argue that such a bias is in fact desirable, as most “real-world” functions are sparse in the Fourier basis (e.g. smooth functions have support mostly on low-frequency irreps). Computationally, low-rankness of matrices is desirable for purposes of both storage and efficient repeated application, so this work could point to low rank approximation schemes in Fourier space for model compression of locally linear G-CNNs. Of course, this argument would need to be made precise, but this is one possible future direction to be analyzed given our results here.
>
> **It would be helpful to have explicit statements about the main contributions/novelty of the paper.**
>
> The main contributions of our paper are characterizing the implicit regularization of arbitrary G-CNNs, including non-abelian and infinite groups, and testing these results empirically both in the realm of our theory (with linear networks) and in cases of more practical interest outside our theory (nonlinear networks, via local linear approximations at randomly sampled points). We refer the reviewer to the abstract as well as to the new paragraph “Our Contributions” on the first page.
>
> **The experiments in the figure make sense to me in the comparison G-CNN vs FC. Both map G-feature maps to scalars and the latter shows that the bias takes place due to architecture and not the data. However, I do not understand the comparison CNN to G-CNN. If the feature maps are functions on G, is a CNN not automatically a G-CNN? Then, what is the difference between the two?**
>
> We apologize for this ambiguous phrasing and have added a clarifying footnote in the main text. Any time we say “CNN”, we mean the most common type of CNN, i.e. an architecture with convolutions with respect to the cyclic (translation) group, whose size is equal to the size of the inputs. In our experiments, for example, a CNN is a G-CNN with G=$Z_n$, a mismatched group relative to the true labeling function. We include CNNs in our plots simply as a common architecture with which to compare.

---

> > ### Author Response · Authors · 2021-11-23
> > **Response to Reviewer nEDA (2/2)**
> >
> > **The closest I get to a description is in caption of Figure 3: “trained … with six isotropic Gaussian data points”. What is an isotropic Gaussian data point?**
> >
> > We thank the reviewer for pointing out this omission. We have included further details in the appendix (section E) on the simulations, including a description of isotropic Gaussian data points as random vectors with entries drawn i.i.d. Standard Normal.
> >
> > **In theorem 5.3 third line: is this a typo? (w.r.t. indexing to l should be l+1?, it says something like Fw_l-Fw_l \geq \lambda)**
> >
> > We thank the reviewer for noting this typo and have fixed it in the revised draft.
> >
> > **The paper could improve a lot from more layman's explanations throughout the paper. For example, it already starts early in the paper with talk about Schatten norms, where to me it is not immediately obvious why these are considered.**
> >
> > In our writing we strove to balance making the statements clear and readable while still including the necessary formal exposition and notation, particularly considering that our primary contributions are theoretical. Regarding the Schatten norm regularization: indeed, this is a surprising and non-obvious result! During the development of this work, the correct form for implicit bias for the non-abelian case was not clear to us until near the end of the KKT-based analysis. We mention it in the beginning to provide a clear sense of our contributions and how they relate to existing work, but have modified the phrasing slightly to better convey that the reader should not be expected to have guessed at this result. We also thank the reviewer for pointing out that we used the term “Schatten norm” before defining it in the notation section, and have included an informal explanation just before Theorem 1, when the term first arises in the main body.
> >
> > To further provide intuition for our results, we have also included in Appendix D clear visualizations of the bias pre- and post-training for each of the networks we study. The sparseness and low-rank bias of the G-CNN in the Fourier regime is readily apparent in this figure.

---

> > ### Comment · Reviewer_nEDA · 2021-11-29
> > **Thank you for the revision**
> >
> > Thank you for the revision and comments. I think this surely improved the quality of the paper. I will stick to my original rating of a 6, as I do not feel confident enough to assess the broader relevance/impact of this work. Overall, I do believe the work is of high quality and is technically sound. As such, I think it is worth publishable.

---

### Official Review · Reviewer_k7aV · 2021-11-04

**Correctness:** 4
**Technical Novelty And Significance:** 2
**Empirical Novelty And Significance:** 1
**Recommendation:** 5
**Confidence:** 4

**Main Review:**

From a technical perspective, the results for abelian groups (as the authors note) follow from appropriately applying the result of Yun et al. (2020). (This is because convolutions in Fourier space can be expressed as pointwise multiplication for abelian groups.) The more interesting technical contribution is for non-abelian groups. In this case, convolutions corresponds to matrix multiplications of matrices which need-not be diagonal, and thus do not correspond to point wise multiplication. For this case, the authors directly analyze the stationary points of the optimization problem.

An appealing aspect of the paper is to highlight that the norm that is implicitly controlled by gradient descent on group equivariant networks takes a simple form, even for general (finite) groups. This technical point is made clearly in the paper, and the paper is generally well-written.

One weakness of the paper is that the it does not explicitly connect the theoretical results with practical settings. First, since group-equivariant convolutional neural networks are not a standard architecture, it could be helpful if the authors added a discussion of what group structures have been used / are meaningful in practice, and provided insight (e.g. in the empirical section) into the implicit regularization of gradient descent in these special cases.

Furthermore, the empirical results are quite limited. Based on Appendix D, the networks appear to be run on very small synthetic datasets (with 6-10 datapoints). For this reason, it is difficult to determine if the takeaways hold for more realistic classification tasks. It would be helpful if the authors ran experiments on standard binary classification datasets (e.g. MNIST or CIFAR).

Lastly, another weakness of this paper is that the results are restricted to single-channel linear networks with full-dimensional kernels. In light of previous work (i.e. Yun et al. (2020, Gunasekar et al. (2018)), the results are somewhat incremental. The paper mainly demonstrates that the 2/L-norm in Gunasekar et al. (2018) for L-layer linear CNNs can be replaced by the (2/L)-Schatten norm. While this norm (especially for non-abelian groups) does give rise to different sparsity structures, the analysis and the conceptual takeaways are somewhat similar those in Gunasekar et al. (2018).


**Summary Of The Paper:**

This paper studies the induced bias of gradient descent on L-layer linear group-equivariant convolutional neural networks. Their main result is: gradient descent implicitly controls the 2/L-Schatten norm of the Fourier transform of the (linear) predictor. More precisely, they show that gradient descent trained on a L-layer group-equivariant CNN with exponential loss on a linearly separable dataset converges in direction to a first-order stationary point of an optimization problem minimizing the 2/L Schatten norm of the Fourier transform of the linear predictor subject to the margin being at least 1.


**Summary Of The Review:**

I recommend weak rejection due to the incremental nature of the technical results in light of previous work, as well as the limitations of the empirical section.

----
Update after author response: I appreciate the additional experiments on the MNIST dataset, and believe that this improves the empirical analysis provided in this paper. However, given the results in previous work (e.g. Gunasekar et al. '18, etc) for the trivial group, I still think that the results in this work are fairly restrictive since they only hold for single-channel linear networks with full-dimensional filters. Although previous work has shown that simple closed-form solutions may not exist in general, it would still be useful to provide insight into how these important parameters affect the implicit bias of gradient descent.

---

> ### Author Response · Authors · 2021-11-23
> **Response to Reviewer k7aV**
>
> We thank the reviewer for their feedback and respond to individual points below.
>
> **Since group-equivariant convolutional neural networks are not a standard architecture, it could be helpful if the authors added a discussion of what group structures have been used / are meaningful in practice, and provided insight (e.g. in the empirical section) into the implicit regularization of gradient descent in these special cases… Furthermore, the empirical results are quite limited. Based on Appendix D, the networks appear to be run on very small synthetic datasets (with 6-10 datapoints). For this reason, it is difficult to determine if the takeaways hold for more realistic classification tasks. It would be helpful if the authors ran experiments on standard binary classification datasets (e.g. MNIST or CIFAR).**
>
> Group structures that are meaningful in practice include translations, rotations (2D or 3D; discrete or continuous), reflections, permutations and subgroups of permutations, and various product groups of all of these operations. Addressing both this point and the reviewer’s suggestion to consider a standard binary classification dataset, we ran additional experiments validating our theory on the MNIST classification dataset under translational, 90 degree rotational, and flip symmetries. These are practical symmetries for many computer vision applications; see e.g. Lafarge et al 2020 and Chidester et al 2019 in medical imaging applications. On both linear and non-linear networks, we find that the $2/L$ Schatten norm remains an implicit regularizer for the G-CNN architecture, but (naturally) not for other architectures: see Figures 4 and 5 in Section 6.
>
> **Lastly, another weakness of this paper is that the results are restricted to single-channel linear networks with full-dimensional kernels. In light of previous work (i.e. Yun et al. (2020, Gunasekar et al. (2018)), the results are somewhat incremental. The paper mainly demonstrates that the 2/L-norm in Gunasekar et al. (2018) for L-layer linear CNNs can be replaced by the (2/L)-Schatten norm. While this norm (especially for non-abelian groups) does give rise to different sparsity structures, the analysis and the conceptual takeaways are somewhat similar those in Gunasekar et al. (2018).**
>
> We thank the reviewer for their feedback, but respectfully disagree regarding the significance of our results. Both our analysis and the resultant regularization differ significantly from that of Gunasekar et al.; we still use KKT conditions, of course, but even formalizing the architecture in an appropriate manner with respect to group Fourier transforms is non-trivial and requires a good deal of care. Working with stacked matrices of varying sizes instead of vectors is also analytically more complex. More broadly, group-equivariant architectures are becoming incredibly widespread for the explicit bias they impose, so we hope our work brings a new perspective to this community in terms of the implicit bias induced by the optimization algorithm.
>
> In addition, our experiments go beyond existing work. By testing a variety of symmetry groups and problem settings, we validate our theoretical predictions even on non-linear networks (via their local linearizations), providing a closer connection to practice. In fact, we have added experiments on the MNIST classification dataset under the group of translational, rotational, and flip symmetries, and find that our theory holds on the linear version of this network; moreover, we evaluated a nonlinear network under the same symmetries using the e2cnn package and local linear approximations, and find that the G-CNN still achieves lower Schatten norm than ordinary CNNs. Please see Figures 4 and 5 in Section 6. To our knowledge, other works in this space have not explored this analysis.
>
> **Lastly, another weakness of this paper is that the results are restricted to single-channel linear networks with full-dimensional kernels.**
>
> The reviewer correctly points out that this single-channel case is beyond the scope of our theory. In fact, as stated in another reviewer’s response, recent work (Jagadeesan et al. 2021) demonstrates that the induced risk, in the 2-layer ordinary convolutional case, is given by an SDP and thus unlikely to be characterized in closed form. Nevertheless, we do believe that analyzing this more complex multi-channel setting would require first understanding the single-channel setting.
>
>
> References:
>
> -Roto-Translation Equivariant Convolutional Networks: Application to Histopathology Image Analysis, Lafarge et al 2020
>
> -Rotation equivariant and invariant neural networks for microscopy image analysis, Chidester et al 2019
>
> -Meena Jagadeesan,  Ilya P. Razenshteyn,  and Suriya Gunasekar.   Inductive bias of multi-channel linear convolutional networks with bounded weight norm. 2021

---

### Official Review · Reviewer_Mfmn · 2021-11-05

**Correctness:** 3
**Technical Novelty And Significance:** 3
**Empirical Novelty And Significance:** 2
**Recommendation:** 6
**Confidence:** 2

**Main Review:**

Strength: Focusing on the non-explicit bias of G-CNN, the results of the analysis are satisfactory.

Weaknesses: I do not see anything special about it.

Questions:
From a theoretical point of view, we would like this kind of result to include the case where the group does not do group convolution as a case where the group is trivial. What does Theorem 1 imply when the group is trivial?\

It would be great if this kind of result could provide some insight in analyzing over parametrized FNN. When we regress the G-equivariant function on an over parametrized FNN, can your results give any insight?\

This result was for finite groups, but there are also convolutions for geometric groups, such as Lie conv. Please tell us about the part of this result where the finite group assumption works and give us some insight into the generalization to Lie groups.

After the revision, the paper was improved, for example by generalization to the Lie group, but in the main the assessment did not change significantly, resulting in the following scores



**Summary Of The Paper:**

Because of the explicit inductive bias of G-CNN, other inductive biases have not been discussed much.
In this paper, the author focuses on and analyzes the non-explicit inductive bias of G-CNNs.
Technically, they present the non-explicit bias in terms of Fourier matrices by using a group Fourier transform, which depends on the group structure.
The results show that learning a linear G-CNN (with linearized β) by gradient descent implicitly biases the singular values of the Fourier matrix coefficients of β to be sparse.
This result has been experimentally confirmed in linear and some nonlinear situations.

**Summary Of The Review:**

Basically, this paper is theoretically well done. The assumption of linearity is a strong one that is far from reality, but it is a reasonable assumption for the current theoretical analysis, and it is confirmed by experiments in the nonlinear case. Overall, I think this is a worthwhile paper.

---

> ### Author Response · Authors · 2021-11-23
> **Response to Reviewer Mfmn (1/2)**
>
> We thank the reviewer for their thoughtful comments and for their appreciation of our theoretical results, and respond to each point in turn.
>
> **From a theoretical point of view, we would like this kind of result to include the case where the group does not do group convolution as a case where the group is trivial. What does Theorem 1 imply when the group is trivial?**
>
> When the group is trivial, the input is one-dimensional and the output of a group convolution with respect to the input is a single scalar value. Subsequent convolutions occur between vectors of length 1, and are therefore just scalar multiplications. Thus, when the group is trivial, our formalism does not make a lot of sense -- it reduces to the case of a diagonal or fully-connected network with one-dimensional inputs and outputs, for instance. In these cases, Gunasekar et al. 2018 implies that the resultant linear predictor (a scalar) has minimal $L_2$ (or equivalently, $L_{2/L}$) norm), which trivially matches our result. However, our result reduces to other simple cases with appropriate choice of G. For example, when the group is a finite cyclic group, a linear G-CNN architecture reduces to a traditional linear CNN architecture, and our implicit bias matches the known implicit bias of linear CNNs toward Fourier sparse solutions. The only case we can imagine where one might set G to be the trivial group is if the architecture has multiple parallel and independent channels. This case is beyond the scope of our theory, and recent work (Jagadeesan et al. 2021) demonstrates that the induced risk, in the 2-layer ordinary convolutional case, is given by an SDP and thus unlikely to be characterized in closed form.
>
> -Meena Jagadeesan,  Ilya P. Razenshteyn,  and Suriya Gunasekar.   Inductive bias of multi-channel linear convolutional networks with bounded weight norm. 2021
>
> **It would be great if this kind of result could provide some insight in analyzing over parametrized FNN. When we regress the G-equivariant function on an over parametrized FNN, can your results give any insight?**
>
> Our experiments indicate that fitting a G-invariant function with an over-parametrized, fully connected (non-invariant) network yields a linear predictor with greater $2/L$ Schatten norm than an over-parametrized G-CNN; see Figures 3, 4, and 5, comparing the “FC” (fully connected) line to the G-CNN line (also see Appendix D visualizing the results). In other words, a fully connected architecture does not exhibit an implicit bias towards the $2/L$ Schatten norm. Generally speaking, we view different architectures as imposing different implicit biases, i.e. different ways of choosing among zero-training-error solutions, and different ground-truth functions as simply changing this feasible set of possible zero-training-error solutions. For instance, recall from Gunasekar et al. 2018 that a fully-connected network has implicit bias given by the $L_2$ norm of the linear predictor $\beta$ in real space. The underlying function only determines the set of feasible predictors (which fit the data), and not this implicit bias. In this sense, the symmetry of the underlying function and the choice of architecture interact only indirectly, and an invariant function fit with a generic (not invariant) architecture does not experience the same implicit bias as if it were fit with a G-CNN.
>
> **This result was for finite groups, but there are also convolutions for geometric groups, such as Lie conv. Please tell us about the part of this result where the finite group assumption works and give us some insight into the generalization to Lie groups.**
>
> The reviewer correctly points out that infinite symmetry groups (such as SO(3)) are also useful in equivariant learning. In short, under a commonly used band-limiting assumption, there is very little difference between the case of Lie groups and finite groups, and our implicit bias results can be easily extended to infinite groups in this case. We have added Corollary 5.5 on this to our submission (with extra details in the Appendix A.2.1), and we thank the reviewer for leading us to realize this simple corollary of our main results.

---

> > ### Author Response · Authors · 2021-11-23
> > **Response to Reviewer Mfmn (2/2)**
> >
> > In more detail: the first difficulty of formally reasoning about an infinite group G lies in the impossibility of storing the output of a convolution with respect to G with finite memory. This is because, in general, convolution yields an arbitrary function on G, which is of course infinite and cannot be stored. In practice, architectures deal with infinite symmetry groups by truncating in Fourier space, which yields neural nets which are very close to, but not exactly, equivariant (depending on whether the input was precisely band-limited, and whether the non-linearities produce functions that remain band-limited). For example, spherical CNNs perform convolutions by truncation and multiplication in Fourier space, and the fully Fourier space Clebsch-Gordan nets similarly band-limit in Fourier space at each layer. Inspired by these works, we consider the case where G is a compact Lie group (which thus is equipped with a unitary Fourier transform) and assume the input data is band-limited and provided in Fourier, rather than real, space. We also assume all filters are band-limited and real-valued, and optimize over these finite non-zero coefficients in Fourier space. Under these assumptions, convolving with respect to G looks like a finite series of matrix multiplications, just as in the finite G case. Thus, the exact same implicit regularization -- namely, the $2/L$ Schatten norm -- holds.

---

> > > ### Comment · Reviewer_Mfmn · 2021-12-02
> > > **Thank you for the revision.**
> > >
> > > Thanks for the reply, I think the paper has been improved after the revision, generalization to the Lie group etc.
> > > However, my assessment of the main aspects did not change, so I keep my score at 6.

---

### Author Response · Authors · 2021-11-23
**Summary of updates and changes**

We thank the reviewers for their thoughtful responses and respond individually below. Here, we summarize our key changes to the submission:

1. To experimentally explore more real-world settings, we performed experiments on learning MNIST data using a fully equivariant nonlinear network via the e2cnn package. Here, we include ReLU activations and global pooling in the final layer to preserve group symmetries. Surprisingly, despite relaxing the assumptions behind our theory, we find similar patterns of implicit bias during training of this network. More details are included in section 6.2.
2. In addition, we verified our theory on a linear G-CNN on MNIST data, as suggested by a reviewer; see section 6.1. (Please note that the figures have been re-numbered relative to the original submission, and some of the original figures have been relegated to the appendix.)
3. We add a straightforward extension of our theoretical results to the case of infinite groups (in particular, compact Lie groups), where gradient descent of the G-CNN is performed in Fourier space over band-limited inputs and band-limited convolutional filters. This case extends our work to include additional symmetry groups of interest in practice, such as continuous rotations. The informal statement is in Corollary 5.5, with details provided in A.2.1 of the Appendix.
4. In response to requests by the reviewers to provide more visual and practical intuition behind our results, we have added to Appendix D visualizations of the implicit bias of the G-CNN in the real and group Fourier regime. Here, one can readily observe that the G-CNN converges towards a solution that is sparse over low-rank irreps in the group Fourier regime, very much in contrast to the fully connected or convolutional networks. Furthermore, one can also observe the consequences of uncertainty principles in the group setting as the G-CNN clearly converges to a rather “dense” solution in the real regime.

---

### Decision · Program_Chairs · 2022-01-20

**Decision:**

Reject

**Comment:**

This paper examines the implicit bias of gradient descent of linear group equivalent convolutional neural networks with a single channel and full-dimensional kernel when trained on separable data with exponential loss. The main result is that the linear predictor converges in direction to the first order stationary point of the minimum 2/L Schatten matrix norm max-margin problem, under some assumptions. This generalizes previous results on linear convolutional neural networks.

I appreciated this paper states the theorems in terms of general group operations; if done correctly and written well, this can be a good reference for future papers. But I think this paper needs a little more work before getting there, as I explain below.

The reviewers were borderline (6,6,6,5) and did not have high confidence. Some stated clarity issues, other criticized the model being used: either that (1) only the case of single channel and full-dimensional kernel we examined, or that (2) the full model is not actually invariant. Given previous results (Jagadeesan et al.) I am OK with (1). I find (2) problematic, but not enough to be a reason to reject the paper.

So I took a closer look.

First, I felt that indeed the paper writing could be improved. Specifically, the notation could be better explained (e.g., the h and g functions in eq. 3 are not defined: what are their range and domain?), and more discussions and examples can be added throughout the paper to clarify the significance of the results.

Second, the experimental results in the non-Abelian case (figure 4a) and non-linear case (figure 5) seemed somewhat weak (not so sparse) after I noticed the y-axis does not start from zero, as in Figure 3a.

Most importantly, looking at the proofs, I felt they were rather incremental, as I explain next. The authors claim their main result, Theorem 5.4, does not follow from Yun et al.'s paper. But Gunsekar et al. 2018b already had KKT condition results for max margin in parameter space, and even stronger results are in [1] (which the authors should cite and discuss clearly). These already give a stronger version of Theorem A.6. So the main extra contribution here is to extend it to a guarantee on function space (in the space of linear functions beta).

But for L>2, I unless I am missing something, I feel this is straightforward, by using results like in [2], where they relate subdifferentials of unitarily invariant matrix functions to the corresponding vector subdifferentials on the singular values (and in the vector case, the subdifferential is trivial).

The L=2 is not trivial as we need to show the condition in Assumption A.7, which is the most technical part of Gunsekar et al. / Yun et al. papers, and is often non-trivial. The authors in this paper, however, did not show this and rather leave to future work in the last paragraph of Appendix A.  I think that this is a concrete opportunity to make the paper better, perhaps following the same methodology in Gunsekar et al. / Yun et al. papers.

Minor comments:
1) The informal Theorem 1 should state we converge to the f.s.p. of eq. 1, not a solution of eq. 1.
2) The main paper is non-searchable, which makes it harder to read.
3) Many hype refs in the appendix do not work well (they get me to some random page).

[1] K. Lyu, and J. Li. "Gradient descent maximizes the margin of homogeneous neural networks." 2019.
[2] A. S. Lewis  The Convex Analysis of Unitarily Invariant Matrix Functions, 1995